# Association between the *GLP1R* A316T Mutation and Adolescent Idiopathic Scoliosis in French Canadian and Italian Cohorts

**DOI:** 10.3390/genes15040481

**Published:** 2024-04-11

**Authors:** Émilie Normand, Anita Franco, Stefan Parent, Giovanni Lombardi, Marco Brayda-Bruno, Alessandra Colombini, Alain Moreau, Valérie Marcil

**Affiliations:** 1Research Center, Sainte-Justine University Hospital Center, Montreal, QC H3T 1C5, Canada; emilie.normand@umontreal.ca; 2Department of Nutrition, Faculty of Medicine, Université de Montréal, Montreal, QC H3T 1A8, Canada; 3Viscogliosi Laboratory in Molecular Genetics of Musculoskeletal Diseases, Research Center, Sainte-Justine University Hospital Center, Montreal, QC H3T 1C5, Canada; anita.franco.hsj@ssss.gouv.qc.ca (A.F.); alain.moreau.hsj@ssss.gouv.qc.ca (A.M.); 4Department of Surgery, Sainte-Justine University Hospital Center, Montreal, QC H3T 1C5, Canada; stefan.parent@umontreal.ca; 5Department of Surgery, Faculty of Medicine, Université de Montréal, Montreal, QC H3C 3J7, Canada; 6Laboratory of Experimental Biochemistry & Molecular Biology, IRCCS Istituto Ortopedico Galeazzi, 20161 Milan, Italy; giovanni.lombardi@grupposandonato.it; 7Department of Athletics, Strength and Conditioning, Poznań University of Physical Education, 61-871 Poznań, Poland; 8Scoliosis Unit, Department of Orthopedics and Traumatology-Spine Surgery III, IRCCS Istituto Ortopedico Galeazzi, 20161 Milan, Italy; marco.brayda@spinecaregroup.it; 9Orthopaedic Biotechnology Lab, IRCCS Istituto Ortopedico Galeazzi, 20161 Milan, Italy; alessandra.colombini@grupposandonato.it; 10Department of Biochemistry and Molecular Medicine, Faculty of Medicine, Université de Montréal, Montreal, QC H3C 3J7, Canada; 11Department of Stomatology, Faculty of Dentistry, Université de Montréal, Montreal, QC H3A 1J4, Canada

**Keywords:** adolescent idiopathic scoliosis, GLP-1R, single-nucleotide polymorphism

## Abstract

Studies have revealed anthropometric discrepancies in girls with adolescent idiopathic scoliosis (AIS) compared to non-scoliotic subjects, such as a higher stature, lower weight, and lower body mass index. While the causes are still unknown, it was proposed that metabolic hormones could play a role in AIS pathophysiology. Our objectives were to evaluate the association of *GLP1R* A316T polymorphism in AIS susceptibility and to study its relationship with disease severity and progression. We performed a retrospective case–control association study with controls and AIS patients from an Italian and French Canadian cohort. The *GLP1R* rs10305492 polymorphism was genotyped in 1025 subjects (313 non-scoliotic controls and 712 AIS patients) using a validated TaqMan allelic discrimination assay. Associations were evaluated by odds ratio and 95% confidence intervals. In the AIS group, there was a higher frequency of the variant genotype A/G (4.2% vs. 1.3%, OR = 3.40, *p* = 0.016) and allele A (2.1% vs. 0.6%, OR = 3.35, *p* = 0.017) than controls. When the AIS group was stratified for severity (≤40° vs. >40°), progression of the disease (progressor vs. non-progressor), curve type, or body mass index, there was no statistically significant difference in the distribution of the polymorphism. Our results support that the *GLP1R* A316T polymorphism is associated with a higher risk of developing AIS, but without being associated with disease severity and progression.

## 1. Introduction

Adolescent idiopathic scoliosis (AIS) is a three-dimensional spinal deformity of unknown cause that develops in peri-pubertal and post-pubertal growing subjects aged 10 to 18 years. It is the most common pediatric spinal deformity with a prevalence of 0.47% to 5.2% in different populations, with females being more frequently and severely affected than males [1,2,3]. The severity of the spinal deformity can be defined by the Cobb angle, ranging from 10° in mild cases to above 40° in severe cases [4].

AIS etiology remains unknown, but numerous hypotheses have been formulated to explain its pathogenesis, which appears to be of multifactorial origin. The proposed etiologic factors for AIS can be grouped into five broad categories: disorders of the central nervous system, growth and bone metabolism, metabolic/hormonal pathways, biomechanics, and genetics [5,6]. Twin and family studies have shown higher rates of concordance between monozygotic twins with AIS in comparison to dizygotic twins, and a higher risk of developing the disease for relatives of AIS patients [7,8,9,10,11]. These findings support a role for genetics in AIS and a high heritability. To date, several genetic variants have been associated with AIS. The identified associated genes are involved in conjunctive tissue structures, growth and puberty, bone formation, metabolism, melatonin pathway, and ciliary functions [12,13,14,15,16,17,18,19,20].

Several studies comparing scoliotic girls to healthy controls have shown lower body weight, body mass index (BMI), and bone mineral density (BMD) in AIS [21,22,23,24]. These anthropometric discrepancies in AIS patients support the notion that the disease disturbs not only normal spinal growth, but also whole-body growth and development. While the etiology of these differences remains unknown, it has been hypothesized that metabolic hormones play a role in the disturbance of body composition and growth in AIS [25,26,27,28,29]. Glucagon-like peptide-1 (GLP-1) is an incretin secreted by intestinal L cells. GLP-1 binds to the glucagon-like peptide-1 receptor (GLP-1R) on the pancreas to stimulate glucose-induced insulin secretion [30]. GLP-1R is a G protein-coupled receptor (GPCR) that is mainly expressed on pancreatic β-cells [31]. GLP-1R activation by GLP-1 plays an important role in energy homeostasis regulation including insulin secretion and biosynthesis, β-cell proliferation, inhibition of glucagon secretion and the control of satiety by reducing appetite. There is also evidence for an implication of GLP-1 and GLP-1R on bone formation and strength [32]. Considering the role of GLP-1 and its receptor in energy and bone metabolism, their malfunction could contribute to the anthropometric differences observed in AIS.

GLP-1R is a seven-transmembrane protein encoded by the human *GLP1R* gene, located on the long arm of chromosome 6 (chr 6p21), and contains 13 exons [33]. The *GLP1R* A316T polymorphism (rs10305492) is a transition substitution (A/G) that has been associated with lower fasting glucose and risk of type 2 diabetes as well as lower insulin response to a glucose challenge [34]. Thus, carriers of the mutation could have an altered insulin response, which could affect their bone metabolism and anthropometric parameters, as observed in AIS.

Based on the relationship between GLP-1R, bone health, and anthropometric features, we performed an association study between the *GLP1R* A316T polymorphism and the susceptibility of AIS in French Canadian and Italian cohorts. Our objectives were to compare the prevalence of the *GLP1R* A316T polymorphism between non-scoliotic control and AIS participants in both cohorts and to study the relationship between this polymorphism and disease severity.

## 2. Materials and Methods

### 2.1. Study and Participants

This is a retrospective case–control study. The AIS patients (9.6–21 years old) included in the French Canadian cohort were recruited prospectively between 2008 and 2013 as part of a multi-center longitudinal study at the Sainte-Justine University Hospital Center, The Montreal Children’s Hospital, and The Shriners Hospital for Children, in Montreal, Canada. Healthy controls (9.1–16.9 years old) were recruited randomly from local schools or were hospital patients undergoing various surgeries.

For the Italian cohort, patients (8.7–18.4 years old) were recruited prospectively between 2008 and 2014 at the IRCCS Istituto Ortopedico Galeazzi, an orthopedic hospital and research institute in Milan, Italy. Healthy controls (8.1–18.7 years old) were recruited randomly from local schools in Milan. In the Italian cohort, controls and AIS patients were met at a single visit in a cross-sectional study. In the French Canadian cohort, controls were also met only once (cross-sectional study), while AIS patients were followed over several visits within the framework of a longitudinal study.

Both male and female individuals were included in the study. All participants were Caucasians of European ancestry. The diagnosis of AIS was established by clinical and radiological examinations. The Cobb angle was assessed on radiographic film and, for patients with multiple curves, the magnitude of the largest one was used in the analysis. Patients with a Cobb angle between 10° to 40° were considered low/moderate cases and patients with Cobb angles > 40° were classified in the severe case group. AIS patients were considered progressive cases when they had a progression of more than 6° at follow-up and/or they had a curve of more than 45° and/or they needed surgery, as suggested by the Scoliosis Research Society guidelines. Curve types were defined as thoracic, lumbar, or thoraco-lumbar with right or left curvatures based on the assessment of an orthopedic surgeon. The most prevalent curve types were “right thoracic-left lumbar” and “right thoracic-left thoraco lumbar”. During the visit, the weight was measured with a calibrated digital scale to the nearest 0.1 kg and height was recorded standing against a wall-mounted stadiometer, to the nearest 0.1 cm. BMI was calculated for each participant (weight(kg)/height(m)^2^). However, data on disease progression, curve type, and BMI were not available in the Italian cohort, so analyses could only be made in the French Canadian one. Controls from both cohorts were also examined by an orthopedic surgeon to rule out any form of scoliosis before enrollment and they were questioned about the family history of scoliosis. Controls with family history of AIS were not eligible in the study.

Informed written consent was obtained from the parents or the legal guardians of all participants, and minors gave their assent. The study in Montreal was approved by the Institutional Review Boards (IRB) of Sainte-Justine University Hospital Center, the Montreal Children’s Hospital, the Shriners Hospital for Children, McGill University, and by the Affluent School Board and The English school Board of Montreal. The study in Milan was approved by the IRB ASL Città di Milano. All experiments were performed following relevant guidelines and human ethics regulations.

### 2.2. Polymorphism Genotyping

Genomic DNA (gDNA) was obtained from peripheral blood and extractions were performed using PureLink^®^ Genomic DNA kit (Thermo Fisher Scientific, Waltham, MA, USA) according to the manufacturer’s protocol. The polymorphism rs10305492 of the *GLP1R* gene (chr6: 39079018, build GRCh38) was genotyped using the TaqMan allelic discrimination assay (C_25615266_30), read with the Applied Biosystems QuantStudio 5 Real-Time PCR Systems and analyzed with Applied Biosystems TaqMan genotyper software (Thermofisher scientific, Waltham, MA, USA, version 1.4.0). On each PCR plate, two standardized negative and positive controls were used to ensure essay validity.

### 2.3. Statistical Analysis

Descriptive data (age, sex, Cobb angle, type of curve, and BMI) were collected and compared with Student’s *t*-test or Chi-square test for categorial variables. Genotypic and allelic frequency of the rs10305492 polymorphism was derived by direct gene counting. The Hardy–Weinberg equilibrium (HWE) test was performed and evaluated by the Chi-square test for goodness-of-fit. Genotypic and allelic frequency was calculated with Chi-square test and crude odds ratio (ORs) and 95% confidence intervals (CIs) were reported. Multiple logistic regression analysis was conducted to characterize the relationship between AIS and the mutation with adjustment for sex and age, separately and combined. Genotypic and allelic frequency was calculated in sub-group according to severity (mild/moderate vs. severe). Also, in the French Canadian cohort, sub-group analyses were performed according to progression (progressor vs. non-progressor) and curve type (right thoracic-left lumbar vs. others and right thoracic-left thoraco lumbar vs. others). BMI (mean ± SD) was compared in AIS participants (French Canadian cohort) carriers vs. non-carriers of the mutation using the *t*-test. Statistical analyses were performed with GraphPad Prism (Boston, MA, USA, version 8.0.1) and SPSS (Chicago, IL, USA, version 27). Statistical significance was set at *p* < 0.05.

## 3. Results

The clinical and demographic data of the 1025 participants included in this study are shown in Table 1. A total of 636 participants were from the French Canadian cohort (*n* = 95 controls and 541 AIS patients) and 389 participants were from the Italian cohort (*n* = 218 controls and 171 AIS patients). The mean age was 12.0 ± 2.5 vs. 14.9 ± 4.5 years old in controls and AIS patients, respectively (*p* < 0.0001). In both cohorts, mean age and proportion of girls were greater in AIS participants than controls. Mean Cobb angle in the AIS group was 31.1 ± 17.8°.

The *GLP1R* rs10305492 polymorphism was in HWE in both groups: controls (*p* = 0.91) and AIS patients (*p* = 0.57). The allelic and genotypic distributions of the polymorphism in AIS patients and healthy controls in the combined French Canadian and Italian cohorts are shown in Table 2. The variant allele and genotype frequencies of *GLP1R* rs10305492 were 0.6% (A), 1.3% (A/G), and 98.7% (G/G) for controls, and 2.1% (A), 4.2% (A/G), and 95.8% (G/G) for AIS patients. The A/G genotype and A allele were associated with a higher risk (3.40- and 3.35-fold, respectively) of AIS prevalence. Including sex as a covariate in the analysis did not change the magnitude of the association between the mutation and AIS [OR: 3.44 (95% CI: 1.13–10.43); *p* = 0.029]. However, despite following the same trend, the associations were no longer statistically significant when the models included age separately [OR: 1.91 (95% CI: 0.53–6.87); *p* = 0.32] and combined with sex [OR: 2.49 (95% CI: 0.58–10.63); *p* = 0.22]

When cohorts were analyzed separately (Appendix A), the same tendency was found, although associations were not statistically significant. In addition, the tendency towards an association between variant allele and genotype frequencies and AIS prevalence was stronger in the French Canadian cohort than in the Italian cohort.

The association of the *GLP1R* rs10305492 polymorphism with the disease severity was also studied and is shown in Table 2. Allele and genotype frequencies were not different between patients with Cobb angles ≤ 40° (low/moderate cases) and those with angles > 40° (severe cases), indicating that the polymorphism is not associated with disease severity.

The association between *GLP1R* rs10305492 polymorphism and the disease progression was assessed in the French Canadian cohort (Table 3). No difference was found in minor allele and genotype frequencies of patients with non-progressive and progressive scoliosis. Similarly, there was no difference when the cohort was analyzed according to curve type, BMI, or when analyzing females only.

## 4. Discussion

This study supports that the *GLP1R* A316T polymorphism is associated with a higher risk of developing AIS in French Canadian and Italian populations. Carriers of this polymorphism were 3.4 times more likely to develop AIS than non-carriers when the two cohorts were studied together with both sexes included in the analysis. Separately, the same trends were identified in both cohorts, but the associations did not reach statistical significance, most likely due to the small sample size, leading to a lack of statistical power. However, the tendency towards an association is stronger in the French Canadian than in the Italian cohort [2.54 (0.70–10.97) vs. 1.28 (0.18–8.96)]. In addition, we found no association between the mutation and disease severity, disease progression, curve type, or BMI.

In both the French Canadian and Italian cohorts, AIS patients were mostly females, which is consistent with the literature reporting that girls are more at risk to develop scoliosis than males [2,3]. Also, the mean age in the AIS group was superior to non-scoliotic controls, which was reflected in the multiple logistic regression analysis. Including age as a covariate led to the loss of statistical significance for the association between the mutation and the disease, although the same trend remained. It is most likely that the impact of age in the analysis is caused by the recruitment process, as all control participants had been evaluated by a spine surgeon and underwent a family history assessment to exclude any suspicion of AIS or other spinal deformities. While it remains an important study limitation, it does not change the overall conclusions regarding the *GLP1R* A316T mutation and AIS.

While participants of the two cohorts are Caucasian of European ancestry, we might have expected differences between groups in the prevalence of the *GLP1R* A316T polymorphism due to the known founder effect in the French Canadian population [35]. In fact, in the Province of Quebec, the history of colonization and its consequences on the population’s genetic heritage impacts the prevalence and/or clinical characteristics of some genetic diseases [36]. Certain genetic diseases are found in specific regions, while others are present across the territory [36]. Examples of the founder effect in the French Canadian population are the high prevalence of myotonic dystrophy in Northeastern Quebec, the autosomal recessive spastic ataxia of Charlevoix-Saguenay, and Tyrosinaemia type 1 found mostly in Saguenay-Lac-Saint-Jean, Charlevoix, and Haute-Côte-Nord [36,37].

The functional impact of the *GLP1R* rs10305492 polymorphism is not yet well-understood. In vitro, the MIN6 β-cells transfected with a mutant plasmid encoding the rs10305492 polymorphism secreted less insulin, had lower cellular cyclic AMP concentration, and had more apoptosis than wild-type cells [38]. These results support a functional impact of the mutation in β cells and a physiological effect on incretin actions. Furthermore, in humans, the presence of the mutation was associated with lower fasting blood glucose and a lower risk of type 2 diabetes [34]. Two hours after a glucose challenge, mutation carriers had higher glucose levels and lower insulin secretion than non-carriers. Conversely, these findings were not replicated by Dorsey-Trevino et al. They investigated the association between the mutation rs10305492, with the incretin response in participants with type 2 diabetes or at risk for developing diabetes [39]. The intervention evaluated the association of the variants with glucose, insulin, and incretin levels. However, they did not find any impact of the mutation on these parameters, most probably because of the small sample size. Hence, more studies are needed to confirm a role of the *GLP1R* A316T polymorphism in insulin secretion, which could possibly, in AIS, influence bone metabolism and growth.

*GLP1R* coding variants, including rs10305492, have been linked to modifications in the protein molecular affinity with its ligands and G proteins (reviewed in [40]). Functional assays on HEK293 cells transfected with the *GLP1R* rs10305492 variant induced an increase in protein Gαs coupling capacity in response to GLP-1 stimulation [41]. Substituting the non-polar residue alanine with a threonine residue in position 316 of the GLP-1R protein influences the central hydrogen bonding network and modifies the orientation of the transmembrane domains 5 and 6, which could lead to a conformation of the intracellular loop 3 associated with enhanced G protein engagement [34]. The in vivo response of the transfected cells with the rs10305492 polymorphism to various GLP-1 agonists (oxyntomodulin, glucagon, semaglutide, and tirzepatide) showed increased protein Gαs recruitment and GLP-1R endocytosis [41]. Moreover, this missense variant caused a loss of hydrophobic interactions and a gain of polar interactions with nearby residues (Asn320 and Gly318), part of the transmembrane binding pocket of the GLP-1R: GLP-1 complex [42]. The response on G-protein coupling capacity of different *GLP1R* mutation assessed in vitro was negatively correlated with participants’ random glucose levels, supporting the clinical observation of lower fasting blood glucose levels in *GLP1R* A316T polymorphism carriers [34].

GLP-1R is a G protein-coupled receptor that can couple to Gαi proteins [31]. Interestingly, it was shown that AIS patients present with a signalling dysfunction of Gαi proteins [43]. In that study, AIS patients were classified in three endophenotype groups according to different alteration of melatonin receptor signalling in AIS osteoblasts. One endophenotype had phosphorylated Gαi1 and Gαi2 that was associated with a higher risk of curve progression compared to the other two groups. It remains to be demonstrated whether the *GLP1R* A316T polymorphism impacts the coupling of the Gαi proteins and its true effect on protein function and AIS pathophysiology.

This study has several limitations. First, the cohort includes twice as many AIS patients than non-scoliotic controls and, following stratification, the sample size of sub-groups used for analysis of disease severity, progression, curve type, and BMI is small. This may have precluded the identification of statistically significant associations. Also, information on the disease progression, curve type, and BMI was not available for the Italian cohort; thus, only participants from the French Canadian cohort were analyzed for these parameters, reducing statistical power. Moreover, while data on treatment outcomes with bracing or surgery were not available, they also could have provided interesting insight on the impact of the mutation during treatment in AIS patients. Finally, given the mean age of control participants, they probably had not all reached skeletal maturity, meaning that an eventual development of scoliosis cannot be ruled out beyond all doubt. While stratifying the cohorts based on growth rate and bone fusion time would have provided additional relevant insights, this information was unfortunately not available. The strengths of the study include its novelty, as the *GLP1R* A316T variant has never been studied in AIS, to our knowledge. Also, two different cohorts were used to investigate the association between the polymorphism and the disease, allowing the study of two genetically similar populations of European descent. Finally, all participants recruited were examined by a spine surgeon, confirming AIS diagnosis and disease severity, but also excluding spinal deformities in the control group.

In summary, our study reveals *GLP1R* as a candidate gene for AIS. Future investigations in larger cohorts from various ethnic groups are required to validate this finding. Mechanistic studies are needed to better understand the impact of the mutation in AIS pathophysiology in relation to energy metabolism.

## Figures and Tables

**Table 1 genes-15-00481-t001:** Demographic data of study participants.

		All		French Canadian Cohort			Italian Cohort		
	Controls	AIS		Controls	AIS		Controls	AIS	
	(*n* = 313)	(*n* = 712)	*p*-Value	(*n* = 95)	(*n* = 541)	*p*-Value	(*n* = 218)	(*n* = 171)	*p*-Value
Sex			<0.0001 ^a^			<0.0001 ^a^			<0.0001 ^a^
number of boys	162.8	106.8		52.2	81.1		111.2	25.6	
number of girls	150.2	605.2		42.8	459.9		106.8	145.4	
Age (years), mean ± SD									
At diagnosis	11.6 ± 2.2	13.8 ± 2.0	<0.0001 ^b^	13.0 ± 2.2	13.8 ± 2.0	0.0003 ^b^	11.0 ± 1.9	13.7 ± 2.2	<0.0001 ^b^
At last follow-up visit	-	NA		-	15.2 ± 2.0	-	-	NA	-
Cobb angle (degrees), mean ± SD	-	31.1 ± 17.8	-	-	29.5 ± 16.9	-	-	35.9 ± 19.7	-

AIS: Adolescent idiopathic scoliosis; NA: Non-applicable. ^a^ Pearson’s Chi-square test (χ^2^). ^b^ Student’s *t*-test.

**Table 2 genes-15-00481-t002:** Association of the *GLP1R* rs10305492 polymorphism with AIS susceptibility and according to Cobb angle severity in both cohorts.

*GLP1R*	Controls	AIS				Low/Moderate Cases ^a^		Severe Cases ^b^		
rs10305492	(*n* = 313)	(*n* = 712)			OR ^c^	(*n* = 531)	OR ^d^	(*n* = 181)	OR ^e^	OR ^f^
	*n* (%)	*n* (%)	χ^2^	*p*-Value	(95% CI)	*n* (%)	(95% CI)	*n* (%)	(95% CI)	(95% CI)
Genotype										
G/G	309 (98.7)	682 (95.8)	5.84	0.016	Reference	508 (95.7)	Reference	174 (96.1)	Reference	Reference
A/G	4 (1.3)	30 (4.2)			3.40 (1.24–9.06)	23 (4.3)	3.50 (1.30–9.46)	7 (3.9)	3.11 (0.98–9.58)	0.89 (0.19–2.02)
Allele										
G	622 (99.4)	1394 (97.9)	5.74	0.017	Reference	1039 (97.9)	Reference	355 (98.1)	Reference	Reference
A	4 (0.6)	30 (2.1)			3.35 (1.24–8.90)	23 (2.1)	3.44 (1.29–9.29)	7 (1.9)	3.07 (0.98–9.42)	0.89 (0.38–1.98)

AIS: Adolescent idiopathic scoliosis, OR: Odds ratio, CI: Confidence interval 95%, Pearson’s Chi-square test (χ^2^). ^a^ Cobb angle 10–40°. ^b^ Cobb angle > 40°. ^c^ OR between controls and AIS patients, ^d^ OR between controls and AIS patients with Cobb ≤ 40° (low/moderate cases), ^e^ OR between controls and AIS patients with Cobb > 40° (severe cases), ^f^ OR between AIS patients with Cobb ≤ 40° (low/moderate cases) and Cobb > 40° (severe cases).

**Table 3 genes-15-00481-t003:** Association of the *GLP1R* rs10305492 polymorphism with AIS susceptibility according to disease progression in the French Canadian cohort.

*GLP1R*	Controls	Non-Progressive Scoliosis ^a^		Progressive Scoliosis ^a^		
rs10305492	(*n* = 95)	(*n* = 354)	OR ^b^	(*n* = 187)	OR ^c^	OR ^d^
	*n* (%)	*n* (%)	(95% CI)	*n* (%)	(95% CI)	(95% CI)
Genotype						
G/G	93 (97.9)	337 (95.2)	Reference	176 (94.1)	Reference	Reference
A/G	2 (2.1)	17 (4.8)	2.35 (0.57–10.40)	11 (5.9)	2.91 (0.69–13.34)	1.24 (0.57–2.70)
Allele						
G	188 (98.9)	691 (97.6)	Reference	352 (97.0)	Reference	Reference
A	2 (1.1)	17 (2.4)	2.31 (0.58–10.19)	11 (3.0)	2.94 (0.71–13.39)	1.27 (0.59–2.68)

AIS: Adolescent idiopathic scoliosis, OR: Odds ratio, CI: Confidence interval. ^a^ Criteria for a progressive scoliosis: curve had increased by 6° or more between two follow-ups, and/or curve reached 45° and/or when surgery was required. ^b^ OR between controls and non-progressive AIS patients. ^c^ OR between controls and progressive AIS patients. ^d^ OR between non-progressive and progressive AIS patients.

## Data Availability

The raw data supporting the conclusions of this article will be made available by the authors on request.

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
