# Peer review of "Association between the GLP1R A316T Mutation and Adolescent Idiopathic Scoliosis in French Canadian and Italian Cohorts"

_genes, 2024, doi:10.3390/genes15040481_

Round 1
Reviewer 1 Report
Comments and Suggestions for Authors
The paper is well written, and provides valuable and novel information that would be of benefit to the larger academic community. Some study design issues were noted and clarity on some aspects were requested. Please see attached file for more in-depth comments.

Author Response
Please see attach file fot the details of the revisions to the manuscript and responses to comments.

Reviewer 2 Report
Comments and Suggestions for Authors
This manuscript aimed to evaluate the association of the GLP1R A316T polymorphism with AIS susceptibility and to investigate its relationship with disease severity and progression.
1. My primary concern regarding this manuscript is that the odds ratios used by the authors to draw their conclusions were derived from crude analyses. The values of the odds ratios and their confidence intervals could be distorted by other confounding effects, such as age and gender, which differed significantly between the case and control groups, as shown in Table 1. These confounders should be adjusted for in the analysis. The data analysis could be redone by a multivariable logistic regression.
2 2. Please revise the following sentence in the abstract: "While the causes are still, it was proposed that metabolic hormones could play a role in AIS pathophysiology."
Author Response
Please see the attach file for the details of the revisions to the manuscript and responses to comments.

Round 2
Reviewer 2 Report
Comments and Suggestions for Authors
I have no more comments.